

# Transition of an estuarine benthic meiofauna assemblage 1.7 and 2.8 years after a mining disaster

Gabriel Coppo[1], Fabiano S. Pais[2], Tiago O. Ferreira[3], Ken M. Halanych[4], Kyle Donnelly[4], Ana Carolina Mazzuco[1] and Angelo F. Bernardino[1]

[1] Grupo de Ecologia Bentônica, Department of Oceanography, Universidade Federal do Espírito Santo, Vitória, Espírito Santo, Brazil
[2] Plataforma de Bioinformática, Instituto René Rachou, FIOCRUZ/Minas, Belo Horizonte, Minas Gerais, Brazil
[3] Escola Superior de Agricultura Luiz de Queiroz, Universidade de São Paulo, Piracicaba, São Paulo, Brazil
[4] Center for Marine Science, University of North Carolina at Wilmington, Wilmington, NC, United States of America

## ABSTRACT

**Background**. Estuaries are transitional coastal ecosystems that are threatened by multiple sources of human pollution. In 2015, mining tailings from an upstream dam failure caused massive metal contamination that impacted benthic assemblages on the Brazilian Rio Doce estuary.

**Methods**. In this study, we investigate and compare meiofaunal assemblages with eDNA metabarcoding 1.7 years (2017) and 2.8 years (2018) after the initial contamination by mine tailings in order to evaluate the continued impact of sediment mine tailing contaminants on the structure of benthic assemblages after the disaster.

**Results**. The community was dominated by Arthropoda and Nematoda 1.7 yr after the impacts (42 and 29% of meiofaunal sequence reads, respectively) but after 2.8 years Arthropoda (64.8% of meiofaunal sequence reads) and Rotifera (11.8%) were the most common taxa. This continued impact on meiofaunal assemblage revealed a lower phylogenetic diversity (7.8-fold) in 2018, despite overall decrease in metal concentration (Al, Ba, Cr, As, Fe, Zn, Mn, Pb, Cd, Co) in sediments. Our data suggests that differences in benthic assemblages and loss of diversity may be influenced by contaminants in sediments of this estuary, and indicate that broad eDNA assessments are greatly useful to understand the full range of biodiversity changes in dynamic estuarine ecosystems.

Corresponding author
Gabriel Coppo, coppogabriel@gmail.com

# INTRODUCTION

Estuaries are considered dynamic and transitional coastal ecosystems with a high variability in environmental conditions. Most of them are highly productive habitats and acts as a nursery for a great diversity of organisms. For this reason, estuaries are considered one of the most valuable ecosystems in the world, providing important ecological services (*Costanza et al., 1997*; *McLeod et al., 2011*; *Pendleton et al., 2012*; *Janakiraman et al., 2017*; *Lana & Bernardino, 2018*). Estuarine environments are naturally stressed and variable habitats due to their plasticity of physic-chemical processes that vary in short spatio-temporal scales
(*e.g.*, changes in salinity and tide) (*Mulik, Sukumaran & Srinivas, 2020*). Nonetheless, during the last century, the contamination of estuarine ecosystems became a worldwide problem (*Irabien et al., 2008*) due to acute and chronic impacts generated by contamination and pollution, which change the composition of animal assemblages closely associated with sedimentary matrix (*Alves et al., 2013*; *Alves et al., 2015*; *Varzim et al., 2019*).

Meiobenthos, or meiofauna, are sediment associated organisms between 50 and 500 μm (*Higgins & Thiel, 1988*; *Meyer, 1990*). Invertebrates larger than 1,000 μm may be included in meiofauna if they spend part of their life as interstitial organisms (*McIntyre, 1969*; *Hakenkamp & Palmer, 2000*). Meiofauna undertake important ecological roles in estuarine ecosystems, through the biomineralization of organic matter and enhancing nutrient regeneration, linking trophic levels of the food web (*Coull, 1999*; *Kennedy & Jacoby, 1999*). Their high sensitivity to anthropogenic inputs make them excellent proxies for estuarine pollution (*Coull, 1999*), and bioindicator for the management of coastal environment (*Ward & Jacoby, 1992*). However, environmental changes in estuaries, caused by human activities, can strongly impact meiofauna community structure and functioning (*Kennedy & Jacoby, 1999*; *Elliott & Quintino, 2007*), often leading to functional and long-term ecological changes (*Gomes et al., 2017*). Salinity, organic matter content and sediment grain size, for example, are strongly related to the spatial distribution of meiofaunal organisms (*Austen & Warwick, 1989*; *Coull, 1999*; *Rutledge & Fleeger, 1993*; *Walters & Bell, 1994*; *Gomes & Bernardino, 2020*).

Due to the difficulty and labor requirements of accurately identifying meiofauna organisms by traditional morphological identification protocols, these organisms are usually neglected in many biodiversity assessments. However, in recent years there have been considerable advances in applying DNA-based methods using metabarcoding techniques to disentangle biodiversity patterns of microorganisms (*Baird & Hajibabaei, 2012*; *Taberlet et al., 2012*), including meiofauna (*Tang, Li & Yan, 2012*; *Faria et al., 2018*; *Fais et al., 2020*). Recent studies have successfully assessed, by environmental DNA (eDNA) metabarcoding, metazoan biodiversity in different marine ecosystems, such as estuaries (*Bernardino et al., 2019*; *Clark et al., 2020*), continental shelf (*Bakker et al., 2019*; *MacNeil et al., 2022*), and coastal sediment (*Aylagas et al., 2018*; *Jeunen et al., 2018*). This approach has proven to be useful in assessing the compositional data from samples containing such organisms, while the eDNA metabarcoding has proven to be a powerful tool to overcome the limitation for meiofaunal morphological identification (*Valentini, Pompanom & Taberlet, 2009*; *Medinger et al., 2010*; *Gielings et al., 2021*).

The use of eDNA to measure and monitor marine and estuarine biodiversity is gaining popularity (*Creer et al., 2010*; *Bik et al., 2012*; *Brannock & Halanych, 2015*; *Brannock et al., 2016*; *Mäechler et al., 2019*; *Ruppert, Kline & Rahman, 2019*; *Berry et al., 2020*; *Clark et al., 2020*; *Naro-Maciel et al., 2022*). Recent metabarcoding studies using eDNA extracted from sediment (*Avó et al., 2017*; *Lanzén et al., 2017*; *Faria et al., 2018*; *Nascimento et al., 2018*; *Bernardino et al., 2019*; *Fais et al., 2020*; *Castro et al., 2021*; *Pawlowski et al., 2022*) demonstrated its usefulness to assess marine biodiversity. For the most part of biodiversity, eDNA metabarcoding can be more efficient than traditional morphological-based taxonomy, enable the bulk identification of multiple species in an environmental sample by

simultaneously amplifying individual DNA barcodes, which can allow the identification of specimens that are small, cryptic or too degraded for morphological identification (*Steyaert et al., 2020*). In addition, it can be an effective technique for determining the quality and recovery in ecosystems following anthropogenic disasters, such as metal contamination after a rupture on a mining dam (*Chariton et al., 2015*; *Cordier et al., 2017*; *Di Battista et al., 2020*; *Martínez et al., 2020*; *He et al., 2021*; *Leasi, Sevigny & Hassett, 2021*).

In point of fact, in November 2015 a large mine tailing dam ruptured in SE Brazil, releasing nearly 50 million m$^3$ of iron ore tailings into the Rio Doce watershed. The mine tailings load was carried over 600 km downstream reaching the Rio Doce estuary and the Atlantic Ocean, where it severely impacted estuarine and coastal ecosystems nearby (*Carmo et al., 2017*; *Queiroz et al., 2018*; *Bernardino et al., 2019*; *Magris et al., 2019*; *Gabriel et al., 2020a*). The tailings, mainly composed of iron oxyhydroxides, were associated to different potentially toxic elements including Mn, Cr, Pb, Hg, As, La, and Sc, which were 24 times higher for Mn (and more than 200 times higher for other metals, such as Zn and Cu) than before the incident (*Queiroz et al., 2018*; *Queiroz et al., 2021*). The first impacts of the tailings deposition in the estuary included loss of several macrofaunal benthic organisms (*Gomes et al., 2017*), contamination of aquatic organisms (*Gabriel et al., 2020a*; *Queiroz et al., 2021*) and changes in sediment bioturbation and biogeochemistry (*Barcellos et al., 2021*; *Queiroz et al., 2021*; *Barcellos et al., 2022*). The mine tailings impacted the benthic macrofauna diversity, composition and trophic groups (*e.g.,* loss of surface-dwelling taxa), and these impacts were still observed on macrofauna even after almost four years (*Gomes et al., 2017*; *Gabriel et al., 2020b*).

eDNA metabarcoding identified effects of this disaster in the meiofaunal assemblages in the Rio Doce estuary in August 2017, 1.7 years after the tailings spill (*Bernardino et al., 2019*). At the time, high levels of Fe contamination were detected in the estuary sediment, suggesting that meiofaunal assemblages were partially influenced by environmental filtering from toxicity of highly contaminated sediments, since this metal concentrations acted as significant predictors of changes in dominant meiofaunal taxa (*e.g.,* nematodes, copepods, ostracods and flatworms) (*Bernardino et al., 2019*). The Fe concentrations significantly increased by two times two days after the impact (*Gomes et al., 2017*), and in August 2017 continued to be 2–20 times higher compared to preserved (Piraquê-Açu-Mirim estuary) or polluted estuaries, such as the Vitória Bay, located in a metropolitan and industrial area approximately 100 km to the south (*Hadlich et al., 2018*). As the time passes and the contamination impacts in Rio Doce are reduced, it is expected that these biological communities will exhibit some degree of recovery, which should be detected by long-term monitoring and biodiversity assessments.

Given the highly dynamic nature of the estuarine ecosystems, and the prediction that levels of contaminants in sediments will decrease with time (see *Gabriel et al., 2021*), we re-evaluated he Rio Doce meiofaunal assemblages 2.8 years (2018) after the initial impact. Our aim was to evaluate the continued impacts on meiofaunal assemblages in response to sediment contamination by metals, through biodiversity assessment and multivariate association. We hypothesized that meiofaunal composition and diversity would be affected by metal concentrations in the impacted estuarine sediments, leading to ecological recovery,

and that higher phylogenetic diversity would occur with a reduction on the contaminant levels.

# MATERIAL & METHODS

## Sampling site and sampling procedures

The Rio Doce estuary (19°38′ to 19°45′S, 39°45′ to 39°55′W; Fig. 1) is located in SE Brazil with a tropical climate, and two well-defined seasons, dry winters (April to September) and wet summer (October to March), and a monthly average rainfall of 145 mm (*Alvares et al., 2013*; *Bernardino et al., 2015*; *Bissoli & Bernardino, 2018*). The estuary is characterized by low salinity levels (0.05–8 ppt) and temperatures between 23.1 and 30.5 °C (*Gomes et al., 2017*; *Bernardino et al., 2018*; *Lana & Bernardino, 2018*; *Gabriel et al., 2021*).

Sampling was carried out in August 2018 at 16 sampling sites distributed throughout the lower portion of the Rio Doce estuary, covering about five km from its mouth (Fig. 1). At each site, we collected two sediment samples (top five cm) using sterile, DNA-free corers and immediately frozen in liquid nitrogen. Additional samples were obtained for determination of grain size, total organic matter and trace metal quantification. All sediment samples were stored in a freezer at −20 °C upon arrival at the laboratory until further analysis. Additionally, water temperature and salinity were measured at each site. Field sampling was approved by SISBIO-IBAMA (sampling license N 24700-1), and data were collected as previously described in *Bernardino et al. (2019)*.

Grain size was determined according to *Suguio (1973)* by sieving and pipetting, and we quantified total organic matter (TOM) gravimetrically by the weight loss after combustion (500 °C for 3 h). Metal concentration in sediment samples was evaluated from two independent replicate samples. For the total trace metal contents, approximately 1 g of the freeze-dried samples was digested by a tri-acid mixture (nine mL of HNO3 + three mL of HF 1 mol/L + five mL of $H_3BO_3$ 5%; *USEPA, 1996*) in a microwave oven digestion system. Vessels containing the samples were shaken and heated at 110 °C for 4 h. After that, we diluted samples to 40 mL in deionized water. We determined the concentrations of trace metals (Al, Ba, Cr, As, Fe, Zn, Mn, Pb, Cd, Co) using aliquots of 0.1mL on an ICP-OES spectrometer (iCAP 6200; Thermo Scientific, Waltham, MA, USA; see *Queiroz et al., 2018*) in triplicate. Standard solutions were prepared from dilution of certified standard solutions and certified reference materials (NIST SRM 2709a) and used for comparison to measured and certified values. Sedimentary and metals concentrations analysis were realized as previously described in *Gabriel et al. (2020a)*.

## DNA extraction and sequencing

Prior to DNA extraction, we elutriated the sediment samples using 45 μm sieves, following the protocol established by *Brannock & Halanych (2015)*, using 950 mL of filtered seawater in a 1L flask, inverting the flask and decanted the liquid over the sieve after the flask was let to sit. After repeating this procedure ten times, sediment retained on the sieve was rinsed to a sterile 50 mL falcon tube, and spun down using an Eppendorf Centrifuge 5430 at room temperature for 3 min at 1,342 × g, and was aliquoted to 20 mL. The sample was mixed using a sterile pipette, and two separate one mL aliquots were

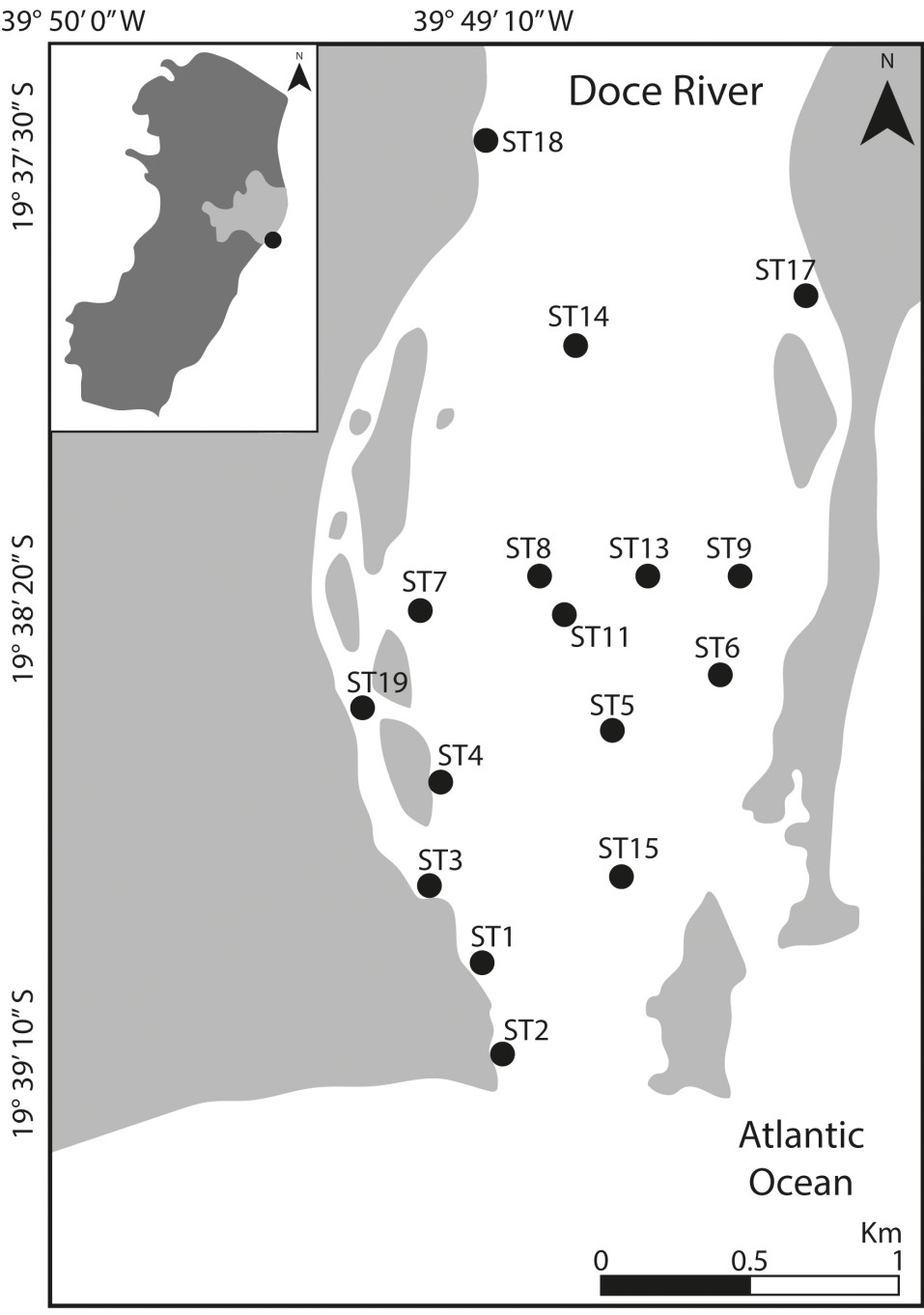

**Figure 1** **Location of the study area.** Map indicating the sampling stations at Rio Doce estuary, on the SE Brazilian coast, in August 2018.

removed and stored in separate sterile 1.5 mL tubes, and stored at −20 °C for DNA extraction. All glassware and materials used during the elutriation process were cleaned, sterilized, and autoclaved between samples. After elutriation, we extracted DNA from the sediment samples using the PowerSoil DNA Isolation® kit (Qiagen) following the

manufacturer's instructions. We verified DNA integrity on a 1% agarose gel and purity (260/230 and 260/280 ratios) using a NanoDrop spectrophotometer (Thermo Fisher Scientific Inc., Waltham, MA, USA). We determined DNA concentration using a Qubit® 4 Fluorometer (Life Technologies-Invitrogen, Carlsbad, CA, USA), and samples were sent to ©NGS Genomic Solutions (Piracicaba, SP, Brazil) for metabarcoding sequencing and construction of the amplicon libraries by HiSeq Illumina platform (2 × 250 bp). The V9 hypervariable region of the 18S SSU rRNA gene was amplified using primers Illumina_Euk_1391f forward primer (GTACACACCGCCCGTC) and Illumina_EukBr reverse primer  (TGATCCTTCTGCAGGTTCACCTAC) (*Medlin et al., 1988*; *Lane, 1991*; *Amaral-Zettler et al., 2008*; *Stoeck et al., 2010*).

## Bioinformatic pipelines

We used the 2021.2 QIIME2 software distribution to process and analyze all demultiplexed raw paired-end reads to estimate the observed taxa (*Bolyen et al., 2018*). Fastq files were first imported as QIIME2 artifacts, and reads were denoised *via* DADA2 (*Callahan et al., 2016*) with the DADA2 *denoise-paired* plugin, setting the p-trunc parameter to 220 to remove low-quality bases, and the p-trim set to 10 to remove primer sequences.

The taxonomic composition of the amplicon sequence variants (ASV), generated after running the DADA2 plugin, were assigned using the machine learning Python library *scikit-learn* (*Pedregosa et al., 2011*). The *feature-classifier* plugin was used to generate the classification results by a pre-trained Naïve Bayes classifier trained on Silva 132 database clustered at 99% similarity (*Quast et al., 2013*), and the taxonomic profiles of each sample were visualized using the *taxa-barplot* plugin. Due to the difference on the number of identified sequences, we normalized datasets from both years to allow analysis and comparison with homogenous sampling depth. We used the 2018 dataset minimum sampling depth (2,282 reads) and resampled each station to the same depth. These filtered/subsampled datasets were used to calculate all diversity metrics.

We reanalyzed and re-identified all sequences from the 2017 assessment realized by *Bernardino et al. (2019)* following this pipeline to guarantee that both datasets (2017 and 2018 assessments) were treated and analyzed using the same techniques and procedures, and to guarantee we were doing a more accurate comparison. Additionally, we built one phylogenetic tree for each dataset using QIIME2. To generate the trees, we used the *align-to-tree-mafft-fasttree* pipeline from the *q2-phylogeny* plugin. After that, we calculated Faith's phylogenetic diversity (PD) using the *diversity core-metrics-phylogenetic* pipeline, based on the phylogenetic tree generated before. The PD is obtained summing the branch lengths on a phylogenetic tree, where longer branches correspond to longer evolutionary times and more distinct taxonomic groups (*Faith, 1992*). Additionally, we plotted rarefaction curves for both assessments. Raw sequence data is deposited in NCBI (SRA: SRR21716030).

## Statistical analysis

Only meiofaunal sequence reads were used for ecological and statistical analysis, and here we considered meiofaunal metazoans the five phyla that are exclusively meiofauna (Gnathostomulida, Kinorhyncha, Loricifera, Gastrotricha, and Tardigrada) and other

metazoans that can be representative of meiofauna during any stage of life and play important role in the sediment (temporary meiofauna) (*Higgins & Thiel, 1988*; *Giere, 2009*). Normality of all environmental data were tested by Shapiro–Wilk test, and when necessary, data were log-transformed (log10 or log10(x + 1)). Differences in environmental variables, phylogenetic diversity, and the relative abundance of taxa between 2017 and 2018 assessments were assessed by Student's *t*-test (*Student, 1908*; *Mann & Whitney, 1947*). The differences on abundance between phyla were analyzed by a One-Way Analysis of Variance (ANOVA), followed by the Tukey *post-hoc* test for multiple comparisons (*Tukey, 1949*; *Underwood, 1997*). A Similarity Percentage Routine (SIMPER) was applied to analyze the contribution of each taxonomic group to the assemblage composition dissimilarity between the two datasets (*Clarke, 1993*). Linear regressions were performed to evaluate the relation between metals concentrations and phylogenetic diversity and, phyla relative abundances. A non-metrical multidimensional scaling (nMDS; *Oksanen et al., 2022*) plot was performed with the meiofaunal assemblage composition in August 2017 and August 2018. A canonical analysis of principal coordinates (CAP; *Anderson & Willis, 2003*) ordination plot was made with the set of environmental variables that better explain the meiofaunal assemblage. Significant differences were defined when $p < 0.05$. All graphical and analytical processes were performed in R environment (*R Core Team, 2022*).

# RESULTS

## Environmental conditions

In the 2018 assessment, the salinity in the estuary at the time of sampling was $0.14 \pm 0.04$, and the temperature ranged from 23.7 °C to 26.3 °C. Sediment grain size of sampled stations indicated a predominance of sand particles (minimum = 48.8% and maximum = 94.1%), and the total organic matter (TOM) varied between 1.5 and 11.8% (Table 1; Table S1). We found a significant decrease in concentration of all measured sediment trace metal compared to the assessment made in 2017 ($p < 0.05$; Table 1; Table S1), except for arsenic which increased ($p = 0.536$; Table 1). We measured an average sediment Fe concentration of 16,566 mg/kg. Associated metals, including As, Cr, and Cd still have showed concentrations above the limits allowed by the current legislation (5.9 mg/kg, 37.3 mg/kg, and 0.6 mg/kg, respectively).

## Assemblage structure and phylogenetic diversity

We reanalyzed the data from the 2017 assessment and found a significantly higher number of meiofaunal sequence reads when compared to the 2018 assessment (2017 = 3,090,870 sequence reads; 2018 = 120,627 meiofaunal sequence reads; $t = 11.147$; $p < 0.001$; Table S3). In the 2017 dataset we identified 12 phyla, which is similar to the 10 phyla identified in the 2018 assessment, with the addition of Micrognathozoa, and Tardigrada. The most frequent phyla in the 2017 assemblages were Arthropoda (41.8%) and Nematoda (29.2%) (Fig. 2A).

We detected a total of 162,330 sequences from the eDNA metabarcoding of Rio Doce estuarine sediments in 2018 (Table S2). After filtering the dataset to remove sequences that were not meiofaunal animals (*e.g.*, bacteria, fungi, algae, protists), we obtained 120,627

**Table 1 Environmental data from sedimentary samples.** Sediment grain size, total organic matter (TOM), and metal concentrations (mg/Kg), as median, minimum and maximum, obtained from sampled station in Rio Doce estuary in August, 2017 and August, 2018. Significant differences ($p < 0.05$) are presented in bold.

| Variables | Year | | | | |
| --- | --- | --- | --- | --- | --- |
| | 2017* | | 2018 | | $p$ |
| | Median | Min - Max | Median | Min–Max | |
| %Sand | 87.8 | 11.8–96.2 | 85.5 | 48.8–94.1 | 0.532 |
| TOM | 3.20 | 1.50–16.8 | 4.00 | 1.50–11.8 | 0.646 |
| Al | 32,495 | 10,066–65,386 | 19,467 | 10,754–27,590 | **<0.001** |
| As | 2.84 | <LQ–53.1 | 4.29 | 0.15–12.6 | 0.536 |
| Ba | 238.7 | 33.3–688.4 | 68.3 | 26.1–177.3 | **<0.001** |
| Cd | 3.25 | 0.57–7.53 | 1.76 | 0.72–2.67 | **<0.001** |
| Co | 9.41 | 3.81–20.9 | 7.18 | 4.78–9.69 | **0.004** |
| Cr | 47.1 | 17.7–79.6 | 25.1 | 10.25–45.3 | **<0.001** |
| Cu | 8.83 | 2.31–16.1 | 4.05 | 0.64–6.65 | **<0.001** |
| Fe | 35,538.3 | 13,204.4–57,923.3 | 15,990.5 | 8,981.7–26,862.1 | **<0.001** |
| Mn | 551.8 | 148.4–1094.9 | 345.3 | 163.5–539.2 | **<0.001** |
| Ni | 14.5 | 7.17–28.6 | 10.1 | 6.27–15.0 | **<0.001** |
| Pb | 101.9 | 4.92–182.2 | 6.52 | 3.68–10.9 | **<0.001** |
| Zn | 35.4 | 15.3–85.9 | 27.4 | 14.6–46.1 | **0.009** |

**Notes.**
*Data from August, 2017 were obtained from *Bernardino et al. (2019)*.

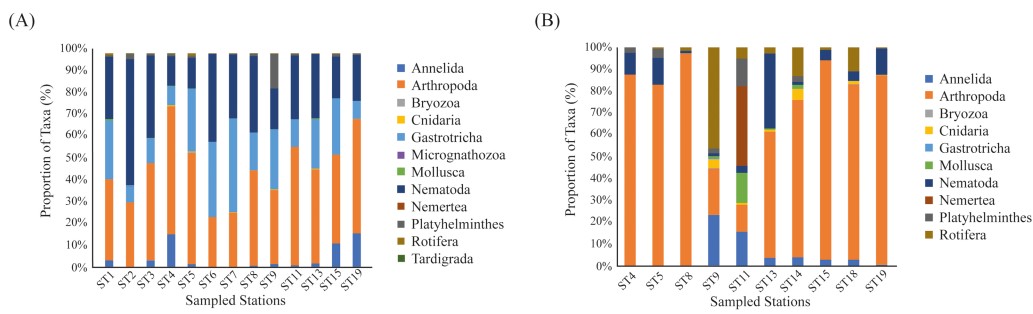

**Figure 2 Frequency of identified taxa.** Barplots showing (A) the proportion of identified Phylum at the Rio Doce estuary in 2017 assessment, and (B) in the 2018 assessment.

sequence reads from ten phyla, most of them identified as Arthropoda (64.8% of sequence reads; Table 2) and Rotifera (11.8%; Table 2). The frequencies were significantly different between phyla ($df = 9$; $F = 12.715$; $p < 0.001$; Fig. 2B; Table S3). The rarefaction curves suggest that the number of meiofaunal taxonomic groups was on overall higher in 2017 when compared to 2018 (Fig. 3).

Further, we observed a significant decrease in phylogenetic diversity (PD) from 2017 to 2018. Meiofaunal assemblages in 2017 had a mean PD of $166.6 \pm 35.1$, when compared to the meiofaunal PD in 2018 of $21.3 \pm 7.2$; a significant decrease in PD of 7.8 times in 2018 ($t = 23.320$, $df = 44$, $p < 0.001$). In addition, we observed the same pattern for Shannon

**Table 2  Frequency of meiofaunal sequences identified in the 2017 and 2018 assessments.** Meiofauna assemblage composition and relative frequency of sequences of each amplicon sequence variants (ASVs) identified at Rio Doce estuary in 2017 and 2018 assessments.

| Phylum | Class | Order | 2017 Assessment | 2018 Assessment |
|---|---|---|---|---|
| Miscellaneous Annelids | | | – | 5.79% |
| | Clitellata | Haplotaxida | 0.21% | 3.09% |
| | | Rhynchobdellida | 0.12% | – |
| | Polychaeta | Echiuroinea | 0.00% | – |
| | | Eunicida | 1.97% | – |
| | | Spionida | 2.73% | 0.13% |
| Miscellaneous Arthropods | | | 0.20% | 19.52% |
| | Arachnida | Acari | 0.34% | 0.09% |
| | Branchiopoda | | – | 0.25% |
| | | Diplostraca | – | <0.01% |
| | Malacostraca | Eucarida | <0.01% | 0.02% |
| | Maxillopoda | | 0.02% | 7.07% |
| | | Calanoida | 0.01% | – |
| | | Hexanauplia(Copepoda) | 0.06% | – |
| | | Cyclopoida | – | 0.57% |
| | | Harpacticoida | 0.10% | 0.13% |
| | Ostracoda | Halocyprida | – | 0.02% |
| | | Podocopida | 33.68% | 25.39% |
| Bryozoa | Gymnolaemata | | 0.03% | – |
| | Phylactolaemata | Plumatellida | – | 0.08% |
| Cnidaria | Anthozoa | Actiniaria | <0.01% | – |
| | | Zoantharia | 0.01% | – |
| | Hydrozoa | | 0.01% | 0.01% |
| | | Anthoathecata | 0.06% | 0.02% |
| | | Limnomedusae | 0.01% | 2.45% |
| | Myxozoa | Bivalvulida | 0.01% | 0.08% |
| Gastrotricha | | Chaetonotida | 23.11% | 0.12% |
| Micrognathozoa | | | <0.01% | – |
| Miscellaneous Molluscs | | | – | <0.01% |
| | Bivalvia | | – | 0.08% |
| | | Myoida | 0.04% | – |
| | | Nuculoida | <0.01% | – |
| | | Veneroida | 0.01% | 2.82% |
| | Gastropoda | Caenogastropoda | 0.06% | – |
| | | Heterobranchia | <0.01% | – |

**Table 2** (*continued*)

| Phylum | Class | Order | 2017 Assessment | 2018 Assessment |
|---|---|---|---|---|
| Nematoda | Chromadorea | | 0.08% | 0.09% |
| | | Aerolaimida | 0.03% | – |
| | | Chromadorida | 0.02% | – |
| | | Desmodorida | 6.43% | 0.23% |
| | | Monhysterida | 16.73% | 4.52% |
| | | Rhabditida | <0.01% | – |
| | | Tylenchida | <0.01% | 0.02% |
| | Enoplea | Dorylaimia | 0.22% | 1.39% |
| | | Enoplida | 6.72% | 0.16% |
| | | Triplonchida | 0.38% | 0.56% |
| Nemertea | Anopla | Heteronemertea | <0.01% | 5.84% |
| | Enopla | Monostilifera | – | 0.09% |
| Miscellaneous Platyhelminthes | | | 0.02% | 0.62% |
| | Catenulida | | 0.01% | 0.31% |
| | Monogenea | Monopisthocotylea | 0.07% | 0.10% |
| | Rhabditophora | Macrostomida | 0.01% | 1.97% |
| | | Proseriata | 0.91% | – |
| | | Rhabdocoela | 5.13% | 0.47% |
| | | Seriata | 0.13% | 0.03% |
| | Trematoda | | – | 0.18% |
| | | Echinostomida | – | 0.01% |
| Rotifera | Bdelloidea | | <0.01% | 14.90% |
| | | Adinetida | – | 0.05% |
| | | Philodinida | – | 0.09% |
| | Monogononta | | <0.01% | 0.60% |
| | | Flosculariacea | 0.01% | – |
| | | Ploimida | 0.23% | 0.04% |
| Tardigrada | Eutardigrada | Parachela | 0.03% | – |

diversity, with significant higher diversity in 2017 (2017 dataset $= 5.46 \pm 0.48$, and 2018 dataset $= 4.75 \pm 0.79$; $df = 21$; $t = 2.639$; $p = 0.015$).

Multivariate analysis revealed significant differences on the composition of meiofauna assemblages in the Rio Doce estuary between years (Fig. 4). The phyla that most contributed to this difference are Nematoda (24%), Gastrotricha (23.3%), and Arthropoda (18.9%); which contributed to 49.25% of the dissimilarity between the 2017 and 2018 assemblages (Table 3).

## Association with metals and sediments

The results of assemblages' composition in 2018 have a negative relation between the Al concentration and the relative abundance of Mollusca ASVs ($F = 4.964$; $R^2 = 0.209$; $p = 0.043$) and Platyhelminthes ASVs ($F = 4.408$; $R^2 = 0.185$; $p = 0.050$). Furthermore, we observed significant negative relation between the Zn concentration ($F = 14.31$; $R^2 = 0.412$;

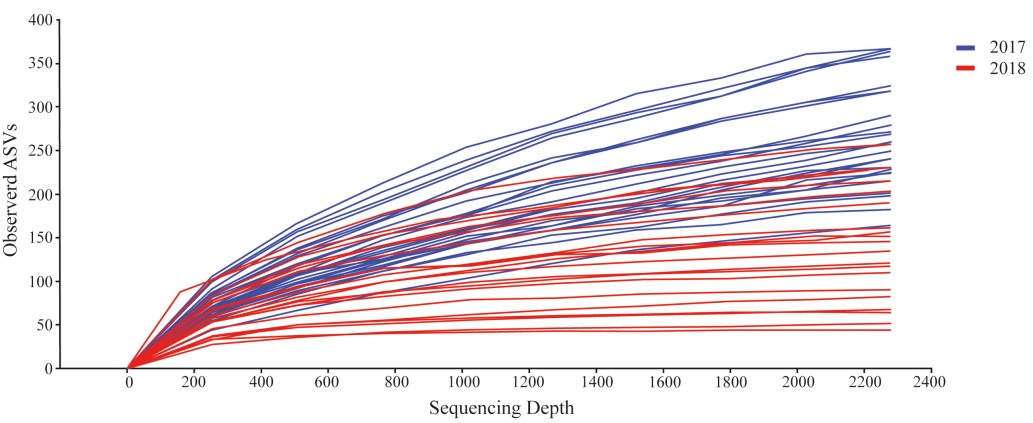

**Figure 3  Rarefaction curves from the datasets from 2017 (blue) and 2018 (red).**

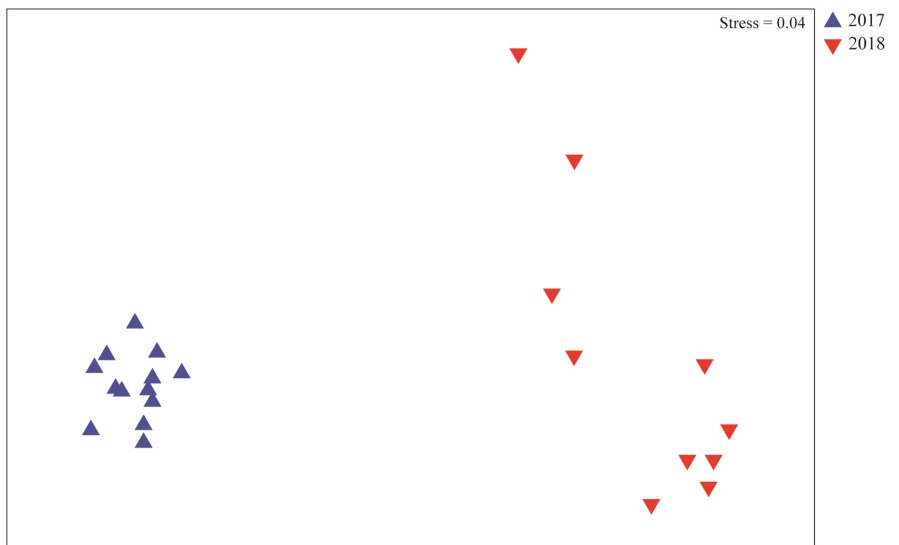

**Figure 4  Meiofaunal assemblages composition.** Non-metrical multidimensional scaling (nMDS) plot based on meiofaunal assemblage composition in August 2017 (blue triangles) and August 2018 (red triangles).

$p = 0.001$), Ni concentration ($F = 9.877$; $R^2 = 0.318$; $p = 0.006$), Pb concentration ($F = 7.302$; $R^2 = 0.249$; $p = 0.015$), Co concentration ($F = 13.11$; $R^2 = 0.389$; $p = 0.002$) and phylogenetic diversity. Even other negative relationships were observed between phyla ASVs and metals concentrations, or between Faith's Phylogenetic Diversity and metals concentrations, they were not significative. The CAP analysis demonstrated that TOM, %Sand, Zn, Cu and Cd is the best set of variables to explain the distribution of meiofaunal

**Table 3  SIMPER results.** Results from similarity percentage analysis (SIMPER) indicating each Phylum contribution to the similarity between 2017 and 2018 assessment in the Rio Doce estuary.

| Phyla | Av. Dissim. | Contrib. (%) | Cumulative (%) |
|---|---|---|---|
| Nematoda | 11.81 | 24 | 24 |
| Gastrotricha | 11.48 | 23.3 | 47.35 |
| Arthropoda | 9.32 | 18.93 | 66.27 |
| Rotifera | 7.71 | 15.65 | 81.92 |
| Nemertea | 2.96 | 6.01 | 87.93 |
| Annelida | 1.99 | 4.04 | 91.97 |
| Mollusca | 1.39 | 2.82 | 94.79 |
| Platyhelminthes | 1.29 | 2.63 | 97.42 |
| Cnidaria | 1.22 | 2.49 | 99.91 |
| Bryozoa | 0.02 | 0.05 | 99.96 |
| Tardigrada | 0.01 | 0.03 | 99.99 |
| Micrognathozoa | 0.00 | 0.01 | 100.00 |

**Notes.**

Av. Dissim., Average Dissimilarity; Contrib., Contribution.

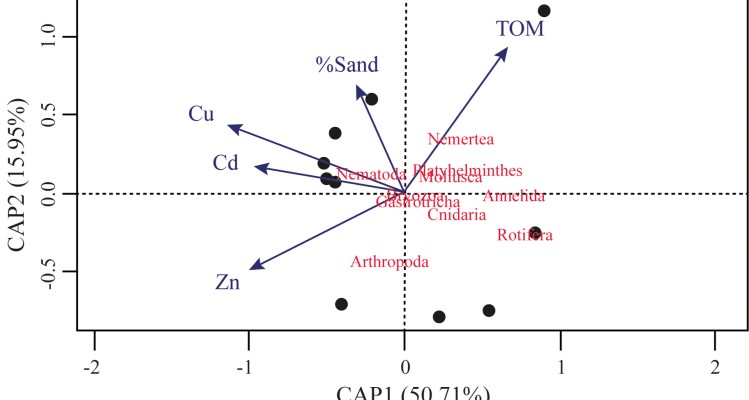

**Figure 5  Canonical analysis of principal coordinates (CAP) ordination of samples according to multivariate distribution of the meiofaunal metazoans identified in the Rio Doce estuary in 2018.**

assemblage in 2018, and this model significatively explain 66.66% of the distribution of the identified meiofaunal metazoans (Fig. 5; $F = 2.378$; $p = 0.044$).

Differences in the composition of assemblages, and in the phylogenetic diversity between 2017 and 2018 can also be observed on the respective phylogenetic trees, built based on the ASVs identified in the dataset of each sampling event. We can observe a more complex, diverse and with longer branches in the tree based on the 2017 assessment (Fig. 6A). In the phylogenetic tree from 2017 the branches are longer and more divided in different nodes, representing more diversity, especially in Nematoda, Gastrotricha and Platyhelminthes. Additionally, in 2018 the meiofaunal assemblage changed, since the branches are shorter

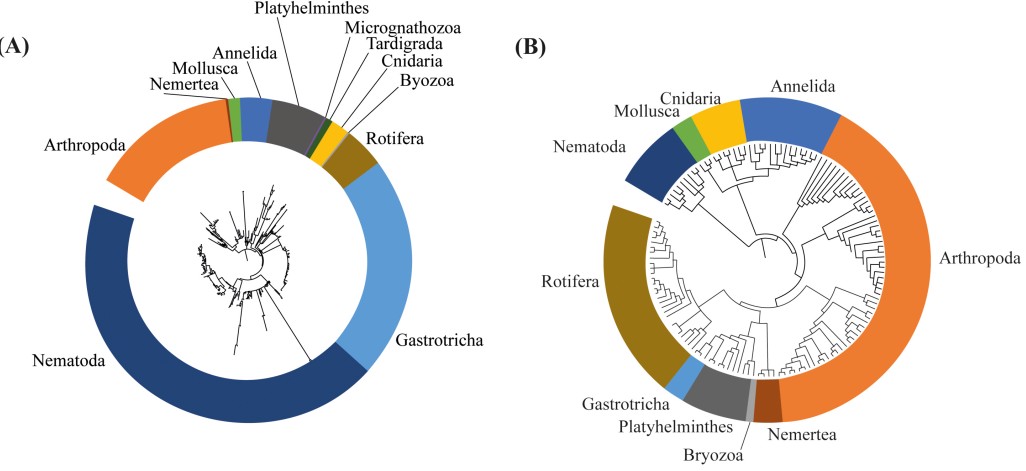

**Figure 6** Phylogenetic trees based on the amplicon sequence variants (ASVs) identified from (A) 2017, and (B) 2018 assessments in the Rio Doce estuary.

and less divided in different nodes. Is notable how Arthropoda and Rotifera become more representative phyla for the assemblage composition (Fig. 6B).

## DISCUSSION

eDNA metabarcoding of the Rio Doce estuary revealed a lower meiofaunal phylogenetic diversity 2.8 years after the mine tailing disaster, which is contrary to our initial hypothesis of a temporal increase of meiofaunal diversity along an expected decrease in sediment contamination. The temporal comparison of meiofauna assemblages showed significant changes in the composition and diversity (Figs. 2; 6) of meiofaunal organisms, which are markedly associated with the metal contamination in the sediments. Therefore, our results support that the meiofaunal assemblage in the Rio Doce estuary has changed substantially between 2017 and 2018, but with observed reductions in phylogenetic diversity, number of sequences, and changes in the relative abundance of each taxon.

Sediment metal concentrations decreased since the initial impacts were observed in the Rio Doce estuary, but concentrations are still well above pre-impact levels (*Gomes et al., 2017*; *Gabriel et al., 2021*). Estuaries are commonly considered ecosystems with low diversity, due to the highly dynamic hydrological conditions (*Gray, Wu & Or, 2002*; *Anila Kumary, 2008*; *Alves et al., 2013*; *Janakiraman et al., 2017*; *Hadlich et al., 2018*). Nematodes and Arthropoda are common taxa in estuarine sediments (*Coull, 1999*; *Dalto & Albuquerque, 2000*), and in the Rio Doce they represented over 70% of the taxa sampled (Table 2; Fig. 2). These taxa were key to differences observed between 2017 and 2018. In 2017, Nematoda was dominant in the same sampled stations representing 29.2% of sequences of meiofauna (*Bernardino et al., 2019*).

Copepods are known as a pollution sensitive taxon (*Won et al., 2018*), but nematodes are highly tolerant to pollution, and some species detoxify absorbed or ingested metals by using metal-binding proteins (*Montserrat et al., 2003*; *Ferraro et al., 2006*). *Millward &*

*Grant (1995)* applied toxicity tests on a nematode community from a severely contaminated estuary, and evidenced that nematodes are resistant to Cu. Thus, the higher dominance of nematodes in 2017 may be related to the higher levels of metals (*Bernardino et al., 2019*); and their decreased abundance in 2018 suggests a temporal succession of dominance; possibly related to a gradual decrease in pollution observed in the estuary (see *Gabriel et al., 2021*). This reduction on the relative abundance of nematodes (from 29.2% to 5.2% of total meiofaunal sequences), which are a potential indicator of contaminated sediments, may indicate an assemblage response to the reduction in the metal concentrations in the sediment, where other less tolerant taxa can compete with taxa that are more tolerant to toxicity.

The significant changes observed in meiofaunal assemblages supports the marked temporal changes in environmental conditions of the estuarine sediments. We additionally observed a stronger degree of dissimilarity in assemblages in 2018, which support high bottom heterogeneity and some recovery. The higher heterogeneity in sediment composition can be a source of species nestedness (or loss) in estuarine sediments (*Menegotto, Dambros & Netto, 2019*), which could explain lower taxonomic diversity and higher dominance of Arthropods in 2018.

The distribution of metals (*e.g.*, Cd, Cu and Zn) may help explain the distribution pattern of meiofaunal metazoans in 2018. *McLeese, Sprague & Ray (1987)* indicated Cd as not toxic at typical environmental concentrations. Some other studies on meiofauna suggest that Cd does not affect species compositions (*Austen & McEvoy, 1997*; *Austen & Somerfield, 1997*). *Trannum et al. (2004)* did not observe negative effects from high concentrations of Cd on the recolonization of different benthic taxa. On the other hand, *Wakkaf et al. (2020)* observed Cd toxicity to meiobenthic nematodes. Copper, a common contaminant in bays and estuaries (*Hadlich et al., 2018*), and considered to be most toxic metal to many marine species (*NAS, 1977*), showed negative correlations with benthic recolonization rates in experiments realized by *Olsgard (1999)* and *Trannum et al. (2004)*. Although Zn is not considered toxic to marine organisms (*Bryan & Langston, 1999*), *Gyedu-Abadio (2011)* found influences of this metal on the structure of nematodes in two estuaries in South Africa.

Metal concentrations had a significant effect on meiofaunal assemblages after 2.8 years, in addition to sedimentary organic content and grain size. Organic matter contents in the sediment plays a key role and is a nutrient source that determine benthic organisms' distribution (*Paarsons, Takahashi & Hargrave, 1984*; *Neto, Bernardino & Netto, 2021*). The distribution of some meiofaunal organisms may be influenced by grain size, like crustaceans that are usually more abundant in coarse sediments (*Tietjen, 1969*; *Hicks & Coull, 1983*). Grain size determines structural and spatial conditions from the habitat, and indirectly influences the physical and chemical parameters of it (*Giere, 2009*). In fact, different studies suggest that abiotic factors, such as grain size and organic matter content, contribute to the patchy distribution of meiofaunal assemblages in a similar pattern observed in the present study (*Nascimento, Karlson & Elmgren, 2008*; *Alves et al., 2009*; *Faria et al., 2018*; *Fais et al., 2020*).

We expected to detect a meiofaunal successional process towards assemblages with higher richness and diversity, when compared to the 2017 assessment, which would suggest a recovery process from chronic impacts of metal contamination. Our study then supports that other factors can influence the rate at which biotic assemblages recover from environmental disasters. Our results suggest that the Rio Doce estuary was not yet on a recovery path after nearly 3 years from the initial impacts, as ecosystems are not considered recovered until a secondary succession returns the ecosystem to the pre-existing condition (*Borja, Dauer & Elliott, 2010*). In this sense, we would need continued long-term assessments to determine its a recovery trajectory (*Latimer et al., 2003*). The recovery of benthic communities can vary greatly from weeks (*Danovaro, Fabiano & Vincx, 1995*) to 10–25 years (*Jones & Schmitz, 2009*; *Borja, Dauer & Elliott, 2010*; *Aderhold et al., 2018*), and some ecosystems may never be technically recovered and end up irreversibly in an alternative state (*Borja, Dauer & Elliott, 2010*). A similar result was observed by *Fleeger et al. (2019)* that did not observe a full recover on meiofaunal assemblages 6.5 years after an oil spill contamination. Our results corroborate those found by *Gambi et al. (2020)* that clearly detected the effect of long-term tailing discharge on benthic diversity after several decades from the end of the mining. In our case, it is difficult or even impossible to determine the state of recovery the Rio Doce estuary since there are no baseline data or long-term studies of meiofaunal assemblages in this estuary.

The meiofaunal phylogenetic diversity from the Rio Doce estuary suggests losses of diversity in assemblage composition from 1.7 to 2.8 years after initial impacts. This may be a result or a response to the chronic effects of the metal concentrations following the disaster since, despite a significant decrease on metal concentrations, the contamination remains above reference values (*Gabriel et al., 2021*). These observed differences in meiofauna assemblages may indicate changes in other biological components, and consequently in the whole estuarine ecosystem. The loss of some meiofauna phyla and the decrease in phylogenetic biodiversity corroborates to this hypothesis.

## CONCLUSION

In conclusion, we observed substantial differences on meiofaunal assemblage composition and diversity in the Rio Doce estuary from 1.7 yrs to 2.8 yrs after a mine tailing disaster. Although sediment metal concentrations decreased in time, we observed fewer identified sequences and phylogenetic diversity. Our results suggest that meiofaunal diversity are now influenced by total organic matter content and grain size, but the continuous contamination by trace metals including Cd, Cu and Zn seems to still influence assemblage diversity. On the other hand, the reduction on Nematoda relative abundance—a tolerant taxa to toxicity—may indicate a recovery of meiofaunal assemblages *via* competition with less tolerant taxa. Additionally, we reinforce that the use of eDNA assessments is very useful and cost-effective to understand the dynamic of estuarine ecosystems and temporal changes on biodiversity. The continued sampling and monitoring on the Rio Doce estuary would be of great importance to understand how this meiofaunal assemblage will respond during the successional process over time.

## ACKNOWLEDGEMENTS

We thank Fabricio Angelo Gabriel for helping with field sampling.

### Funding

This work was funded by PELD and PRONEM grants from Fundação de Amparo do Espirito Santo (790548684/2017, 84532092/2018), Coordenação de Aperfeiçoamento de Pessoal em Nível Superior CAPES and Conselho Nacional de Pesquisa e Desenvolvimento CNPq (441243/2016-9; 441107/2020-6) to AFB. Gabriel C. Coppo received a scholarship from Coordenação de Aperfeiçoamento de Pessoal em Nível Superior CAPES, and a travel grant from Fundação de Amparo do Espirito Santo (393/2021). The funders had no role in study design, data collection and analysis, decision to publish, or preparation of the manuscript.

### Grant Disclosures

The following grant information was disclosed by the authors:
Fundação de Amparo do Espirito Santo: 790548684/2017, 84532092/2018.
Coordenação de Aperfeiçoamento de Pessoal em Nível Superior CAPES.
Conselho Nacional de Pesquisa e Desenvolvimento CNPq: 441243/2016-9, 441107/2020-6.
Coordenação de Aperfeiçoamento de Pessoal em Nível Superior CAPES.
Fundação de Amparo do Espirito Santo:  393/2021.

### Competing Interests

Angelo F. Bernardino is an Academic Editor for PeerJ.

### Author Contributions

- Gabriel Coppo conceived and designed the experiments, performed the experiments, analyzed the data, prepared figures and/or tables, authored or reviewed drafts of the article, and approved the final draft.
- Fabiano S. Pais performed the experiments, analyzed the data, authored or reviewed drafts of the article, contributed reagents/materials/analysis tools, and approved the final draft.
- Tiago O. Ferreira performed the experiments, analyzed the data, authored or reviewed drafts of the article, contributed reagents/materials/analysis tools, and approved the final draft.
- Ken M. Halanych performed the experiments, analyzed the data, authored or reviewed drafts of the article, contributed reagents/materials/analysis tools, and approved the final draft.
- Kyle Donnelly performed the experiments, analyzed the data, authored or reviewed drafts of the article, and approved the final draft.
- Ana Carolina Mazzuco conceived and designed the experiments, performed the experiments, analyzed the data, prepared figures and/or tables, authored or reviewed drafts of the article, and approved the final draft.

- Angelo F. Bernardino conceived and designed the experiments, performed the experiments, analyzed the data, prepared figures and/or tables, authored or reviewed drafts of the article, contributed reagents/materials/analysis tools, and approved the final draft.

## Field Study Permissions

The following information was supplied relating to field study approvals (i.e., approving body and any reference numbers):

Field sampling was approved by the SISBIO-IBAMA (sampling license N 24700-1).

## DNA Deposition

The following information was supplied regarding the deposition of DNA sequences:

The sequences are available at NCBI SRA: SRR21716030, SAMN31026701.

The RioDoce18ST data is also available at NCBI: SRR21716030.

The data is also available at FigShare: Coppo, Gabriel (2022): riodoce18. figshare. Dataset. https://doi.org/10.6084/m9.figshare.21133567.v1.

https://trace.ncbi.nlm.nih.gov/Traces/?view=run_browser&page_size=10&acc=SRR21716030&display=download.

## Data Availability

The raw data is available in the Supplementary Files and Table 1.

## Supplemental Information

Supplemental information for this article can be found online at http://dx.doi.org/10.7717/peerj.14992#supplemental-information.

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
