# Peer review of "Transition of an estuarine benthic meiofauna assemblage 1.7 and 2.8 years after a mining disaster"

_PeerJ, doi:10.7717/peerj.14992_

## Round 0.1 · original submission · Major Revisions

I have received three independent reviews of your study. While all reviewers clearly recognised the quality/novelty of your work, they have collectively raised a number of issues that will need to be addressed in your revised manuscript.

In particular, it was highlighted that English needs another round of solid proof editing/restructuring, that Figures need to be uploaded in a high-resolution format, and that the bioinformatics pipeline, and in particular calculation of Faith PD, need clarification.

Overall, the reviewers have provided you with excellent suggestions on how to improve the manuscript, and I be looking forward to receiving your revised manuscript along with a point-by-point response to their comments.

With warm regards,
Xavier

Reviewer 1 ·

Basic reporting

This manuscript by Coppo and colleagues compares 18S eDNA metabarcoding results from sediments of an estuary heavily impacted by metal contamination from mine tailings in 2015. Samples were taken 1.7 and 2.8 years after the impact, with very different community composition between the two time points.

While parts of the manuscript are relatively clear, it is often difficult to follow the methodology as well as the results due to unclear language. The section on bioinformatic “pipelines” in particular needs to be clarified, as well as the Results. Further, the aim of the study, as described in L109-114 is not sufficiently clear as it is now written. For example, please revise what “compare temporal changes in meiofaunal assemblages [...]” mean. Do you mean to contrast the temporal changes in between the two years? I think it may be helpful to relate the hypothesis that then follows to ecological recovery and the decrease of metal contamination that has taken place.

In particular, it is not clear how Faith’s PD was calculated using QIIME2. How was the tree used for this metric made and the alignments on which it was based? Note also that amplicons of partial rRNA sequences are not appropriate for de novo calculation of trees, although it may be possible to calculate Faith’s PD based on taxonomic position in e.g. SILVA using QIIME2. Why were two trees, one per dataset, calcualted, when the ASVs resulted from a common analysis of the two datasets? It is rather the case that members found at the end of some branches may not be present in one dataset that causes the difference in Faith’s PD. (Also please change “arms” to “branches” throuhgout the manuscript)

Another important ceveat with the analysis is that Faith’s PD is heavily influenced by sequencing depth and the authors (if I understand this paragraph correctly) did note a significant difference in sequencing depth between the two datasets, with 2017 data having a higher sequencing depth (L217-219). Thus, it is not appropriate to use the difference in this metric to contrast the two datasets as it may not reflect a true difference in phylogenetic alpha diversity, but rather a bias of the different sequencing depth. This is also true for several comparisons using sequencing read counts (“representative sequences”) between taxa. I suggest that a different alpha diversity metric with less sensitivity to differences in sequencing depth, such as Shannon diversity, is used to estimate diversity. Another option is to use random subsampling of the individual samples or the two datasets before calculating Faith’s PD. In any case, comparisons between the abundance of taxa in 2017 v 2018 also have to be carried out using relative abundances or subsampled data in order to be meaningful. Table 3 should use such counts instead of the number of reads. Further, it is not clear what percentages of “representative amplicon sequence variants” mean. Are these statements, e.g. in the Abstract, about comparisons of abundance or the number of ASVs, i.e. richness? Note that richness is directly influenced by sequencing depth also and has to be compensated by subsampling, if possible repeated random subsampling, in order to calculate a “rarefied” richness.

The figures appear in very low resolution, making them difficult to interpret. I am not sure if they were uploaded with higher resolution and automatically downsampled, but if not, this is an important consideration before resubmitting.

It would improve the manuscript to mention more about the severity of the disturbance on the macroinvertebrate community, in the Introduction or the Discussion, if such results are also available from other studies.

Finally, it is not stated if the raw sequence data is deposited in any public repository.

Experimental design

See comments regarding Faith's PD stated above since they are related to basic reporting (lack of clarity).

Further, was any effort made to remove contaminants (human or animal foodstuff DNA etc) as well as DNA from dead organisms such as terrestrial insects? It should be relatively straightforward to remove such taxa in many cases based on taxonomic classification and it will, in addition to the different read depth, have a large influence on Faith’s PD. If this was done, please describe it and if not, please attempt to do this before reanalysing the data.

Validity of the findings

With the current lack of clarity and the problem of how Faiths PD was calculated, it is not clear to me if the results are correctly interpreted or not, but I would look forward to review a revised manuscript where an alternative method is used and it is more clearly described what constitutes differences in (relative) abundance of taxa or vs. differences in richness (number of ASVs).

Additional comments

SPECIFIC COMMENTS

L28: Something is missing before “2.8 years”, proabably “after”.

L44: Suggest rephrasing to “Acute and chronic human impacts”. Now “impacts” is repeated.

L56: It is true that estuaries are subjected to “natural stressors” due to e.g. tidal dynamics and rapid changes in salinity, but I am not aware of “ecosystem natural dynamics” strongly impacting the ecommunities in the sense that it can lead to “long term ecological changes”. This would imply an important shift due to natural reasons, and I am not sure if it is what you mean. The cited references do not mention this neither as far as I can tell and I suggest that “natural dynamics” is not mentioned as this is not related to the aim of the study in any case.

L60 “typos” is likely a typo for “type” but it the sentence is not clear even with this cahnge and it is not clear what then is meant with “indicate community types”.

L65: please remove “diversity” from “DNA-based diversity methods”

L76: Suggest to rephrase “is being increasingly reported” to “is gaining popularity” or similar to improve readability.

L79: Lanzén et al. 2017 is missing from the references

L94: Please rephrase “were associates to”

L95: Please clarify what was 24 times higher compared to before the accident. Mean sediment metal concentrations in the estuary, and if so, in the same stations, and for what metal, or for the sum of the metals?

L99-100: The effects 1.7 years after the accident can hardly be called “initial”. This part of the introduction is important, since it is so central to the interpretation of the results. It could be that initial disturbance was higher and is not recovering with dropping metal concentrations. But it could also be the case that there is a lag in the biological effect of metal contamination and that the initial effects were still getting stronger after 2.5 years, in spite of the fact that metal concentrations in the sediment have started to decrease. Please clarify and extend this section briefly.

L122: 8% salinity would be hypersaline brine and above the 3.5% of seawater. Do you mean ppt (i.e. g/L or “salinity units”)?

L161: what does “estimate the observed taxa” mean here? Observed using morphology? Or simply that QIIME2 was used to process and analyse the metabarcoding data? Please clarify.

L170: Change “de” to “the” or similar

L175: please reprhase “no biased by methods temporal comparison”. The meaning is not clear to me

L180: which parameters were found to differ significantly from a normal distribution and did you repeat the Shapiro-Wilk test after log-transformation. Please also consider to use a nonparametric test instead of Students t, such as e.g. Wilcoxon ranked test.

L186-187: Please rephrase ”differences [...] assessments were analysed” to “differences [..] were assessed” (by student’s t)

L192: I fail to understand what the “phyla that contributed most to the dissimilarity between the asessments” mean and how they were identified.

L193 and Ñ245: Please refrain from stating that regressions were evaluating cause-effect relations. Regressions can evaluate statistical correlation but never the cause and effect, which is always left to interpretation and other observations.

L199: “Significantive” -> “significant”


L207: Refer to Table 1 and supplementary data.

L210: There is no need to write out standard deviation (?) and minimum and maximum and it would be more meaningful to compare the two between 2017 and 2018.

L218: Why are only the number of reads for one dataset written when the point is to compare them. Please include both and clarify what “n(small ) = 10, n(big) =13” means? Is that a typo?

L221: It is unclear what “the most represenatives” means here. Please rephrase. Also see general comments above about this.

L220 and L225-7: Do you mean that 0.2% of the reads were unassigned at phylum level? 46 taxa before or after removing non-meiofaunal taxa? In L220, 12 vs. 10 phyla are mentioned so this is not consistent. Also, did you remove contaminants such as human or food DNA and trace DNA from insects (see comment above)?

L227-232; It is not clear what “frequencies” refers to here and if it is actually relative abundance or richness (number of unique ASVs). In any case, a comparison between absolute read counts of taxa without compensating for the significant difference in sequencing depth is inappropriate (see above comment also).

L238: What does “ASV’s diveristy” reer to here?

L242 and Table 3: How was the percent of dissimilarity contributed by each phylum calculated? It does not seem from Figure 2 that the overall difference in phylum rank abundances was 92%.

L272: I disagree with the statement that the authors partly refute a hypothesis regarding that the communities would be direcrly impacted by metals. This is not the hypothesis put forward in the Introduction and there is no evidence for it. To evaluate it, a baseline community before the metal contamination would be needed. In any case the opposite is clear, i.e. that the community has changed substantially between 2017 and 2018 according to the data presented and I cannot see how it can be ruled out that this was caused by a change in metal concentration.

Reviewer 2 ·

Basic reporting

Dear authors and editors,
I reviewed this article, named "Temporal changes on an estuarine benthic meiofauna assemblage 1.7 and 2.8 years after a mining disaster" with interest. I am pleased that the meiofauna continues to be considered as a bioindicator and that metagonomics studies are being applied to this purpose. I thank the authors for their contribution. Although the MS is of interest to me and I believe it can be for the entire scientific community, there are some tricks to be done before the actual publication.
First of all, in some places the English is not very academic, indeed it's a very street-art form. I would suggest a reading by a qualified person before future submission, with the use of adverbs and syntax more suited to scientific English, rather than discursive one. The literature is perhaps too exhaustive and old in some points of the document: even in this case, a reconsideration is needed.
In general, the structure of the article is acceptable; the data has been shared but the figures lack quality. I would obviously suggest the inclusion of the sequences in the official database (e.g., SRA).
No excessive criticism on the hypotheses considered. However, a small clarification should be made in the MS on:
- which taxonomic groups were actually considered meiofauna by the authors (do they correspond to what was actually found or in Table 2 some important taxonomic groups for meiofauna are missing)?
- why the choice of the V9 and the primer pairs listed?

Experimental design

Again, in general there are no errors or serious considerations to be made. Further comments in section 4.

Validity of the findings

Again, in general there are no errors or serious considerations to be made. Further comments in section 4.

Additional comments

Please, look at these comments to improve your MS:

Line 39: “and acts as a nursery” (the subject in the sentence is “the most of” not the “estuaries”).
Line 40. “…of organisms. For this reason, estuaries are…”.
Line 42. “Nonetheless, during the last century, the contamination of estuarine ecosystems became a worldwide problem (Irabien et al., 2008) due to acute and chronic impacts generated by contamination and pollution, which change the composition of animal assemblages closely associated with sedimentary matrix” --> I would suggest to insert only the most relevant and/or recent: a lot of references for an intro (Ferris and Ferris, 1979; Moore and Bett, 1989; Coull, 1999; Monserrat et al., 2003; Alves et al., 2013; Alves et al., 2015).
Line 50. McIntyre (https://doi.org/10.1111/j.1469-185X.1969.tb00828.x ) should be the first to have talk about “interstitial meiofauna”. Please check this reference and change/add if necessary.
Line 53-54. Moreover, Ward and Jacoby considered meiofauna as bioindicator in the management of coastal environment (https://doi.org/10.1016/0025-326X(92)90220-Z)
Line 55-56. As you said, both natural and anthropogenic causes are stressors in the estuarine. However, in the beginning of the intro you didn’t talk about the natural stressors. I’d suggest to write a very small sentence about it in the first paragraph.
Line 60-61. The spatial distribution of meiofauna was discussed also by metabarcoding in these papers too: e.g., https://doi.org/10.1016/j.marenvres.2018.06.013; https://doi.org/10.1016/j.ecss.2020.106683; https://doi.org/10.5194/osd-9-1853-2012. You should already insert some references about it in this part of the ms.
Line 66-67. Here it seems that you say that these two references speak of metabarcoding on meiofauna. Readjust the sentence.
Line 79. 2015 isn’t so recent (7 years from now). Nascimento 2018 (10.1038/s41598-018-30179-1), Fais (2020 10.1016/j.ecss.2020.106683; 10.1016/j.ecss.2020.106683), Pawlowski (10.1016/j.scitotenv.2021.151783) and Castro (10.1007/s10750-021-04600-2) are most recent.
Line 88-89. I really don’t like this detachment from talking about the advantages of DNA metabarcoding for meiofauna at the introduction of your work. A link is missing here.
Line 126-127. Did you take sediments from corers, Falcon tubes or what/how?
Line 131. I suppose is “grain size”.
Line 145 etc. what is the amount of sediment used for extraction? talk about elutriation but this can be good for an exhaustive amount of sediment and not for small quantities. PowerSoil, actually, is for samples up to 2.50g. Moreover, why did you choose the V9 region and these primer pair? So, be more specific in this important part of the ms.
Line 175-176: “Also”? A more professional grammar is required.
Line 183-184: Which taxa have you considered "meiofauna only". You did not say whether you used a conservationist definition (therefore, based on mesh-size or dry weight) or a broader one (therefore, based on the functional diversity of the community). It would be appropriate to give a list, somehow.
Line 204. Salinity is typically very low, as expressed by you. Can you give an explanation for this? Simple curiosity
Line 225-232. Some protists are effectively meiofauna and not all the Arthropoda are. Again, therefore, it is urgent to define what you are looking for. After this, I’d suggest to re-check ASVs following this list, coz it’s possible you were introducing biases here. it's probably something you did, given Table 2, but it doesn't show through in the material and methods in the text.
Line 362: Sampling is an important part of scientific research. Truthfully, excluding such students from the ms names is sad. At the very least, give their specific names in the acknowledgements once your article is ready for publication.
Figures are in a very bad quality.

Reviewer 3 ·

Basic reporting

Figures are highly pixilated

Experimental design

Methods requires some editing - see my longer comment below.

Validity of the findings

The results are currently reported as total counts and would be better suited as relative abundances. This may alter the findings.

Additional comments

In this study, the authors use 18S amplicons as a tool to measure recovery of the benthic eukaryote assemblage to mining tailings between 2017 and 2018. The approach is interesting and has identified structural changes that align with shifts in metal contamination between these two years. I have provided comments below to aid the authors with clear delivery of these results. I do not envisage that these changes will alter the findings substantially however the reduction in nematodes between years may signal benthic recovery contradicting the final concluding paragraphs.


Figures are highly pixilated. Likely caused during the upload process.
“Transition of an …” or “Recovery of an…” may be more appropriate that temporal within the title. Temporal datasets are multi time-point assessments and although these two samples were taken at roughly the same time within the year, there are many factors that are not accounted for that may impact a temporal assessment.
Abstract – “temporal succession”. To test temporal succession this study would require many more time points. Please consider replacing this (e.g. “to evaluate the continued/legacy impact of sediment mine tailing contaminants on the structure of benthic assemblages” - or something that describes that the study is investigating the contaminants that are stored within the sediments.
Methods
L 145-148. To aid the reader please explain what elutriation is in more detail. It is unclear if the filtrate or the sediment was used in the extraction.
L176-178 how this was achieved. Faith 1992 utilises a tree. To my knowledge it does not build one. MUSCLE or MAFFT followed by Fasttree is once approach. I am not familiar with what is built into QIIME2.
L182 - General comment – the two dataset vary significantly in sequencing depth. 2018 is 3.7% of the depth provided by 2017. There is no report of this analysis in the results. It is difficult to interpret the findings if the 2017 data has not been subsampled to match the 2018 data. Or the provision of rationale as to why these reads were kept.
Results
L204 – 207 Are you referring to 2017 or 2018 or both years? How many samples are your averages representing?
L212 – 213 What are the current limits? It would aid interpretation if the number was listed.
L207 How much variation in “and the total organic matter (TOM) varied between 1.5 and 11.8%.” is associated with the sampling gradient versus the temporal sampling? Does this section only refer to 2018? Please add the year.
L214-216 The site location numbering is confusing in relation to the data interpretation. Is it possible to renumber/rename your samples to match a gradient… Such as distance from ocean or organic carbon.. or contamination level? I’m confused what the current naming reflects. This is especially relevant when making sense of why ST13 is the lowest read count versus ST14 having the highest.
L215 how many total reads?
L219 I do not understand what this refers to “representative sequences; t = 11.147, n(small) = 10, n(big) = 13, p < 0.001; Table S3).”
L223-232 and L215-2016 are coupled. It would aid readability if these sections were merged.
L224-225 It is not clear why the results reports the number of phyla prior to filtering.
L228 to 232 These counts would be more informative as % of total reads. This comment also refers to Table 2. By reporting abundances the data takes into account the differences in sequencing depth. With relative abundances the 2018 data is represented by a dominance of 3 Arthropod orders and a single Rotifer. There are 14 orders that have relative abundances greater than 1%. However, the prominent representation is in the phylum Arthropoda. In comparison, 2017 is represented by arthropods, Gastrotricha, and nematodes with 9 orders with relative abundances greater than 1%. From this assessment diversity has decreased however is there an association with the presence of these taxa and their tolerances to high metal concentrations? The type of nematode can be reflective of higher contaminations levels. The reduction in the diversity of this group could suggest that these sediments are recovering. Pivoting “recovery” within this study is complex as there are no pre-disturbance samples.
L235 At was level of grouping was the PD calculated
L238 How was the rarefraction curve calculated. Was this with or without replacement.
L240-243 – These results are represented as % whereas L228-232 are counts. It is not possible to compare years without consistency.
Discussion
The results suggest that the diversity of nematode – a potential indicator of contaminated sediments – deceases between 2017 to 2018. This would support a community recovering from the impacts of toxicity via competition with less tolerant taxa. I agree that eDNA and its application within this study are powerful for monitoring environmental assemblages. However, I do not agree that a decrease in diversity is associated with continuous contamination.

---

## Round 0.2 · Minor Revisions

Dear Dr Coppo and co-authors,

I have received three review assessment of your revised manuscript. While two reviewer now accept your manuscript, reviewer#1 still has a number of major concerns with statistical analyses, particularly with regards to imbalance in sequencing depth between the 2017 and 2018 datasets. I encourage you to carefully address these remaining issues which will be sent back to reviewer#1 for final assessment.

With warm regards,
Xavier

Reviewer 1 ·

Basic reporting

Most of my initial concerns have been addressed in this revised manuscript. However, there are still some things that are not clear after the revisions. First of all, my comment about how Faith’s PD is heavily affected by sequencing depth has not been taken into account, which is problematic due to the much higher sequencing depth of the 2017 dataset. I appreciate the assurance that Shannon diversity was still higher in the 2017 dataset after random subsampling (which is my interpretation of “filtering it by maximum sequencing depth” as written in the response to reviewers). However, I cannot see that this is reported in the manuscript itself. Before I can consider that this manuscript fulfills the requirements of appropriate analysis and interpretation, I would like to see random (1) random subsampling of the individual samples to a homogenous sequencing depth, or similar, (2) the inclusion of the same number of samples from the same locations in both the 2017 and 2018 datasets and finally (3) using these filtered / subsampled datasets for the calculation of Faith’s PD and Shannon diversity.

I also fail to understand why the correlation analysis between metal concentrations and the abundance of specific taxa was limited to the 2018 dataset only. Please motivate this or repeat it with all samples including the 2017 dataset, which should provide better statistical significance and more robust results.

Table 2: The reported sum of each major group is often smaller than the sum of its constituent taxa. For example relative abundance of Arhtropoda is reported as 0.2% in 2017 and 19.5% in 2018. Further, this does not correspond to the abundances reported in Results. Please correct or explain this.

Table 3: I appreciate the inclusion of this table. However, the columns 2017 and 2018 “mean” are not clear. If this is the total number of sequencing reads, again, then that would imply that the analysis has not been done using relative abundances as stated.

Experimental design

see Basic Reporting

Validity of the findings

see Basic Reporting

Additional comments

L204: Here, it appears that the raw counts of sequences were still used rather than relative abundances (“Differences in [...], and the number of sequences between 2017 and 2018 assessments were assessed by Student’s t-test”). This does not correspond to what is reported in the Results.

L205 and elsewhere: Please use “relative abundances of taxa”, “abundances” or similar rather than the more ambiguous term “frequency of sequences”, throughout the manuscript:

L107–110: Environmental filtering refers to how the environment (typically physicochemical parameters rather than the biological communities) shapes the biological communities. Here, high levels of Fe in the sediments are speculated to have led to the disappearance of parts of the meiofaunal community, which is likely. However, the high levels of Fe themselves cannot be considered “suggesting that meiofaunal assemblages were partially influenced by environmental filtering [..]”. You have to provide an example of how the community has changed to be able to suggest that. Please rephrase this sentence accordingly. Also provide a reference regarding the approximate Fe levels before the disaster to put the high levels in context. If no such measures exist, you could refer to normal levels in similar habitats. Is the Fe concentration 10x or 1000x higher than typical levels, for example, or is the change more subtle? Were there any other evident signs like red discolouring from iron oxide?

L167–168: Please also mention the reverse primer position as you have for the forward one.

L220: Missing “the” before “2018”.

L230: change “significant higher” to “significantly higher”.

L231 and elsewhere: please use “sequence reads” or similar instead of the more ambigous “representative sequences” throughout the manuscript. The later can easily be confused with richness, i.e. the number of unique sequences representing each taxon.

L241–244: The relative abundaces of Arthropoda and Rotifera were already reported just above which makes this unnecessarily repetitive and harder to read. I would remove these lines completely and instead just refer to Table 2 above in L235, reporting the top two or three taxa as already done.

Reviewer 2 ·

Basic reporting

I had the opportunity to review the manuscript by Coppo et al, titled "Transition of an estuarine benthic meiofauna assemblage 1.7 and 2.8 years after a mining disaster".
First of all, I commend all the authors for their efforts in accepting the suggestions of the reviewers in a polite and judicious manner. I assure you it is not for everyone. Thanks again for having given a further contribution to the world of meiofauna and metabarcoding.

I consider the document adequately sufficient to be published now, following the suggestions provided and the relative corrections. An effort has been made to improve the English, both in form and content. The references have been changed and updated as suggested. They adequately uploaded figures and the raw data can be found on the online database. The part of materials and methods deficient has been sufficiently improved.

Experimental design

Now, the scope and the methodology are well described.

Validity of the findings

Now, the research in well defined and relevant for the knowledge on meiofauna. believe that this research is an important contribution in the ongoing debate on the use of meiofauna as an indicator of ecosystem status. The use of metabarcoding can undoubtedly increase this potential and this kind research could serve to change the mind on the political decisions on official bioindicators to be used.

Additional comments

I just have a little note. The sentence in line 102-103 that you added, however, seems to me strange between the two paragraphs. What if you wrote "In addition, it can be an effective technique for determining the quality and recovery in ecosystems following anthropogenic disasters, such as metal contamination after a rupture on a mining dam (References...). In point of fact, in November 2015..."

Reviewer 3 ·

Basic reporting

no comment

Experimental design

no comment

Validity of the findings

no comment

Additional comments

I thank the authors for the time they have invested in improving their manuscript and addressing my concerns.

My only comment is that the use of "besides" within the text is not common usage.

---

## Round 0.3 · accepted · Accept

I am pleased to accept this revised manuscript for publication in PeerJ - Congratulations!

I also take the opportunity to thank the reviewers for their valuable contribution in improving this work.

With warm regards,
Xavier

Reviewer 1 ·

Basic reporting

All my previous concerns have been addressed and I now consider the manuscript appropriate for publication.

Experimental design

All my previous concerns have been addressed and I now consider the manuscript appropriate for publication.

Validity of the findings

All my previous concerns have been addressed and I now consider the manuscript appropriate for publication.